# Impact of Integrated Genetic Information on Diagnosis and Prognostication for Myeloproliferative Neoplasms in the Next-Generation Sequencing Era

**DOI:** 10.3390/jcm10051033

**Published:** 2021-03-03

**Authors:** Jong-Mi Lee, Howon Lee, Ki-Seong Eom, Sung-Eun Lee, Myungshin Kim, Yonggoo Kim

**Affiliations:** 1Department of Laboratory Medicine, College of Medicine, The Catholic University of Korea, Seoul 06591, Korea; jongmi1226@catholic.ac.kr (J.-M.L.); navyshow@naver.com (H.L.); 2Catholic Genetic Laboratory Center, Seoul St. Mary’s Hospital, College of Medicine, The Catholic University of Korea, Seoul 06591, Korea; 3Department of Hematology, Seoul St. Mary’s Hematology Hospital, College of Medicine, The Catholic University of Korea, Seoul 06591, Korea; dreom@catholic.ac.kr (K.-S.E.); lee86@catholic.ac.kr (S.-E.L.)

**Keywords:** next generation sequencing, myeloproliferative neoplasm, diagnosis, prognosis, risk stratification

## Abstract

Since next-generation sequencing has been widely used in clinical laboratories, the diagnosis and risk stratification of hematologic malignancies are greatly dependent on genetic aberrations. In this study, we analyzed the genomic landscapes of 200 patients with myeloproliferative neoplasms (MPNs) and evaluated the impact of the genomic landscape on diagnosis and risk stratification. Mutations in *JAK2*, *CALR* and *MPL* were detected in 76.4% of MPNs. The proportion of patients with clonal genetic markers increased up to 86.4% when all detectable genetic aberrations were included. Significant co-occurring genetic aberrations potentially associated with phenotype and/or disease progression, including those in *JAK2/SF3B1* and *TP53*/del(13q), del(5q), −7/del(7q) and complex karyotypes, were detected. We also identified genetic aberrations associated with patient outcomes: *TP53* and −7/del(7q) were associated with an inferior chance of survival, *RUNX1*, *TP53* and *IDH1/2* were associated with leukemic transformation and *SF3B1*, *IDH1/2*, *ASXL1* and del(20q) were associated with fibrotic progression. We compared risk stratification systems and found that mutation-enhanced prognostic scoring systems could identify lower risk polycythemia vera, essential thrombocythemia and higher risk primary myelofibrosis. Furthermore, the new risk stratification systems showed a better predictive capacity for patient outcome. These results collectively indicate that integrated genetic information can enhance diagnosis and prognostication in patients with myeloproliferative neoplasms.

## 1. Introduction

Myeloproliferative neoplasms (MPNs) are characterized by the clonal proliferation of hematopoietic cells that are fully differentiated and functional. MPNs are mainly classified into polycythemia vera (PV), essential thrombocythemia (ET) and primary myelofibrosis (PMF), according to disease manifestations. Some rare disease categories, such as chronic neutrophilic leukemia (CNL) and chronic eosinophilic leukemia, are also considered MPNs, in addition to those MPNs that are unclassified. The diagnosis of MPN and the distinction of its disease categories are based on blood cell counts, bone marrow (BM) morphology and molecular testing. In the 2016 WHO classification, the importance of molecular markers was increasing compared to that in the previous version. Mutations in three well-known driver genes (*JAK2*, *CALR*, and *MPL*) are used to diagnose PV, ET and PMF. Recently, *CSF3R* was identified as a driver gene in CNL [1,2]. *JAK2* mutations have been detected in 90% to 95% of PV patients and 50% to 60% of ET and PMF patients [3,4,5]. *CALR* mutations are the second most common mutation in MPNs, and are found in 20% to 25% of ET and 25% to 30% of PMF patients [6,7,8,9]. *MPL* mutations are found in 3% of ET patients and 5% of PMF patients [10,11]. Patients without mutations in any of these driver genes can be diagnosed with ET or PMF when their clinical and hematologic features meet specific criteria, which includes clonal mutations in any of the following genes: *ASXL1*, *EZH2*, *TET2*, *IDH1/2*, *SRSF2*, and *SF3B1* [1,9]. Knowledge of the mutational landscape not only facilitates the objective diagnosis of MPN, but also can potentially be used as a prognostic factor. This area of research has benefited from the application of next-generation sequencing (NGS) technology in clinical laboratories as well as research laboratories. Important gene mutations associated with patient outcomes, such as mutations in *TP53* and *ASXL1*, have been identified using NGS approaches [12,13,14,15,16].

The risk stratification of MPNs is based on clinical and hematologic features. Advanced age, leukocytosis and venous thrombosis are considered to be risk factors for survival in PV [17] and ET [18]. PMF-specific risk models, such as the International Prognostic Scoring System (IPSS) [19], Dynamic IPSS (DIPSS) [20] and DIPSS-plus (which includes cytogenetic abnormalities in addition to DIPSS), have been developed [21]. The results of recent studies indicate that gene mutations are significant prognostic factors in MPN, and mutation-enhanced prognostic systems for MPNs have been introduced [22,23,24]. These systems can be used to determine the risk for survival as well as disease progression, including leukemic transformation and fibrotic progression [25,26]. Moreover, the integration of genetic information with clinical variables can enable the prediction of patient outcomes and allow for the development of personalized therapeutic plans based on the consideration of causal biologic mechanisms [27,28,29]. It is challenging (at least in the present moment) because data are too limited to make a consensus across ethnicities and countries under different clinical environments.

In this study, we analyzed genetic aberrations using cytogenetic and molecular genetic methods (including NGS) to reveal their impact on the diagnosis of MPNs in the current era. We also tried to evaluate their contributions to risk stratification for survival as well as disease progression.

## 2. Materials and Methods

### 2.1. Patients and Samples

Among patients diagnosed with MPN and treated at Seoul St. Mary’s Hematology Hospital, a total of 200 patients who requested an NGS study or available DNA samples for NGS analysis were included in this study. MPN diagnoses and their subtypes were strictly re-evaluated and classified based on the 2017 WHO classifications [1]. Demographics and clinical and hematologic features, including outcome data, were reviewed retrospectively from medical records. Most DNA samples were from BM aspirates taken at the time of diagnosis, but some were collected at the time of symptom aggregation during observation or over the course of treatment.

### 2.2. Molecular Analysis

DNA was extracted using the QIAamp DNA Mini Kit (Qiagen, Hilden, Germany). DNA concentration and purity were assessed using an ND-1000 spectrophotometer (Nanodrop Technologies, Wilmington, DE, USA). NGS was performed using a customized myeloid panel (the “SM panel”), as described in our previous report [30]. In short, the SM panel contains 87 genes frequently found mutated in patients with MPN (Appendix A). Target capturing sequencing was performed using a customized target kit (3039061, Agilent Technologies, Santa Clara, CA, USA) according to the manufacturer’s instructions. DNA libraries were constructed according to the protocol provided by the manufacturer and sequencing was performed on an Illumina HiSeq4000 platform (Illumina, Inc., San Diego, CA, USA).

Sequenced reads were mapped to the human reference genome (hg19, Genome Reference Consortium, February 2009) using the Burrows–Wheeler aligner. To call variants, we used VarScan v.2.3.9 with mpileup2snp and mpileup2indel. SnpEff v.4.2 was used to select variants located in coding sequences and predict their functional consequences. Annotated variants were further classified into four tiers according to the standards and guidelines of the Association for Molecular Pathology (AMP) [31]. All variants with a minor allele frequency >0.01 were filtered out based on the Exome Aggregation Consortium (ExAC, http://exac.broadinstitute.org/ (accessed on 23 April 2020)) and genome aggregation database (gnomAD, https://gnomad.broadinstitute.org/ (accessed on 23 April 2020)), as well as the ethnicity-specific Korean Variant Archive (KOVA, http://kobic.re.kr/kova/ (accessed on 23 April 2020)). Variants of synonymous, intronic or noncoding regions were further excluded. Among the remaining variants, variants with more than 20 reads and 5% variant allele frequencies (VAF) were considered to be mutations. Canonical mutations in *JAK2*, *CALR*, and *MPL* fewer than 5% VAF were considered to be present with a low allele burden. All mutations were manually verified using the Integrative Genomic Viewer.

### 2.3. Cytogenetic Study

A total of 169 BM karyotypes were available. Conventional BM karyotyping was performed on G-banded metaphase chromosomes using routine techniques. Karyotypes were interpreted according to ISCN 2016 [32]. The number of cytogenetic events was considered to be the sum of chromosome gains, losses, partial gains, partial losses and rearrangements.

### 2.4. Statistical Analyses

Group comparisons according to phenotype were performed using the Kruskal–Wallis test and Chi-square test. A Mann–Whitney test and Fisher’s exact test were also used for post-hoc analysis. The significance of the co-occurrences of mutations and chromosomal abnormalities was determined using Fisher’s exact test.

The Chi-square test was used to assess whether the distributions of risk scores from two systems were systematically different. We defined overall survival (OS) as the time from diagnosis to death from any cause, and event-free survival (EFS) as the time from diagnosis to leukemic transformation, myelofibrosis progression or death. Medical record tracking was done until February 2020. The Kaplan–Meier Log–rank test was used to estimate OS and EFS in each group. Cox proportional hazards regression was used for univariate and multivariate analyses. Statistical analyses were performed using MedCalc version 19.1.7 (MedCalc software, Mariakerke, Belgium).

## 3. Results

### 3.1. Genetic Landscape of MPNs

The patients’ clinical and hematologic characteristics are summarized in Table 1. According to the disease category of WHO classification, ET was the largest group (*n* = 70, 35.0%), followed by PMF (*n* = 66, 33.0%), PV (*n* = 55, 27.5%) and other MPNs (*n* = 9, 4.5%) (consisting of chronic neutrophilic leukemia (CNL, *n* = 4, 2.0%) and MPN-unclassifiable diseases (*n* = 5, 2.5%)). Age at diagnosis differed significantly according to the disease category. ET patients (43.2 ± 15.3 years) were younger than PV (50.3 ± 12.3 years), PMF (52.5 ± 13.6 years) and other MPN (61.4 ± 14.2 years) patients (*p* = 0.0034, *p* = 0.0002 and *p* = 0.0024, respectively). Clinical and genetic variables are described in Appendix A according to their respective disease category.

The genetic landscapes for all patients are shown in Figure 1A. Mutations were identified in 85.5% of patients (*n* = 171). The mean number of mutations was 1.3 ± 1.0, and ET patients harbored significantly fewer mutations than PMF patients (1.1 ± 0.9 vs. 1.6 ± 1.1, *p* = 0.024). *JAK2* mutations (*n* = 98) were the most common, followed by *CALR*, *ASXL1*, *TET2* and *MPL* mutations (*n* = 43, 24, 17 and 5, respectively). Abnormal karyotypes were identified in 22.5% of patients (38/169), and 4.7% (8/169) of individuals had a complex karyotype with three or more cytogenetic abnormalities. The proportion of abnormal karyotypes in PMF patients (37.5%, 21/56) was higher than that in PV (18.6%, 8/43) or ET (9.8%, 6/61) patients (*p* = 0.047 and *p* = 0.0004, respectively). −5/del(5q) (*n* = 10) was the most common cytogenetic abnormality, followed by del(20q) (*n* = 9), del(13q) (*n* = 9) and −7/del(7q) (*n* = 6) (Figure 1B). We also analyzed the co-occurrence of mutations and abnormal karyotypes. Multiple pairs of co-occurring mutations were found, including *JAK2* with *SF3B1*, *ASXL1* with *SRSF2*, and *DNMT3A* with *IDH1/2* (*p* = 0.017, *p* = 0.01 and *p* = 0.012, respectively). *TP53* mutations were significantly associated with del(13q), −5/del(5q), −7/del(7q) and complex karyotypes (*p* = 0.038, *p* < 0.001, *p* < 0.001 and *p* < 0.001, respectively) (Figure 1C) [33]. We reviewed the Prussian blue iron staining of the available BM samples with *JAK2/SF3B1* co-mutations and found ring sideroblasts in all three samples. Based on the results of the genetic landscapes (including mutations and abnormal karyotypes), we further investigated the associations between genetic aberrations and diagnoses of MPNs.

### 3.2. Impact of Genetic Aberrations on Diagnosis

*JAK2* (*n* = 98, 49.0%), *CALR* (*n* = 43, 21.5%) and *MPL* (*n* = 5, 2.5%) mutations were detected in 76.4% of PV, ET and PMF patients (*n* = 146). *JAK2* mutations were detected in 49 PV patients (89.1%) while *JAK2*, *CARL* and *MPL* mutations were detected in 26 (37.1%), 22 (31.4%) and 3 (4.3%) ET patients, respectively. *JAK2*, *CARL* and *MPL* mutations were detected in 23 (34.8%), 21 (31.8%), and 2 (3.0%) PMF patients, respectively. Among 39 triple-negative ET and PMF patients, eight (two ET and six PMF) patients had myeloid neoplasm-associated mutations in *ASXL1*, *EZH2*, *TET2*, *IDH1/2*, *SRSF2* or *SF3B1*. *CSF3R* T618I mutation was detected in two of four CNL patients. Thus, clonal genetic markers for diagnosis according the WHO classification were present in 89.1% of PV patients, 75.7% of ET patients, 78.8% of PMF patients and 50% of CNL patients, representing 80.6% (*n* = 154) of all MPN patients.

We considered all gene mutations, including those not mentioned above, in addition to an abnormal karyotype, as clonal genetic markers. Genetic aberrations were detected in 12 patients, including mutations of *U2AF1*, *DNMT3A*, *RUNX1*, *TP53*, *ZRSR2*, *KDM2B*, *NRAS* and *KIT* genes, as well as karyotype abnormalities such as del(20q) and -5/del(5q). The proportion of MPN patients with clonal genetic markers increased up to 86.4% (*n* = 165). When broken down according to MPN type, 90.9% of PV patients, 77.1% of ET patients and 92.4% of PMF patients had clonal genetic markers (Table 2). We found that implementing all detectable genetic aberrations strengthened diagnosis, especially for cases without mutations in the three main MPN drivers.

### 3.3. Prognostic Impact of Genetic Aberrations

We then evaluated the impact of genetic aberrations on prognosis. The mean overall survival of the study population was 400.5 months, with a 95% confidence interval (CI) of 367.0-434.1 months. Fifteen deaths were recorded in the whole series (2 ET, 8 PMF and 5 other MPN). The clinical outcome in terms of the shortest OS was significantly poorer in other MPN group (33.6 months, 95% CI, 19.7–47.5) than it was in the PV group (325.0 months, 95% CI, 325.0–325.0), ET group (296.4 months, 95% CI, 272.0–320.7) or PMF group (318.8 months, 95% CI, 218.8–418.7) (Figure 2A). Among nine patients in other MPN groups, five died during follow-up. Patients with more genetic aberrations (≥2) showed poorer outcomes (*p* = 0.041). Then, we analyzed the prognostic impact of genetic aberrations in patients with PV, ET and PMF. Univariate analysis showed that mutations in *TP53*, −5/del(5q), −7/del(7q), del(20q), del(13q), the number of abnormal karyotypes, and a complex karyotype were associated with poor outcomes. The diagnosis of PMF and the presence of bone marrow fibrosis were also associated with poor outcomes. Hemoglobin level and *JAK2* mutation were associated with a favorable outcome (Figure 2B). Next, we investigated the impact of clinical and hematologic variables and genetic aberrations on disease progression, including leukemic transformation and fibrotic progression. Nine patients experienced leukemic transformation (two PV, three ET and four PMF patients), and there was no difference in frequency between the three disease groups. *IDH1/2*, *RUNX1* and *TP53* mutations, −5/del(5q), −7/del(7q), the number of abnormal karyotypes and a complex karyotype were significant risk factors for leukemic transformation (Figure 2C). Fibrotic progression occurred in 24 (8 PV and 16 ET) patients. *ASXL1*, *IDH1/2*, and *SF3B1* mutations, the number of mutations and del(20q) were associated with fibrotic progression (Figure 2D). Details of the prognostic impact of each parameter are summarized in Appendix A.

In addition, we performed multivariate analysis. *TP53* mutation, −7/del(7q) and a diagnosis of PMF were adverse survival factors. Peripheral blood blast counts, in addition to *RUNX1*, *IDH1/2* and *TP53* mutations, were identified as risk factors for leukemic transformation. *ASXL1*, *SF3B1* and *IDH1/2* mutations, the number of mutations and del(20q) were defined as risk factors for fibrotic progression (Table 3). We also investigated the prognostic impact of genetic aberrations in each disease category. Despite the limited numbers of patients in each disease category and duration, genetic aberrations identified as risk factors by multivariate analysis had statistical significance in log rank analysis (Appendix A). These results collectively highlighted genetic aberrations associated with outcome, such as inferior survival, leukemic transformation and fibrotic progression.

### 3.4. Impact of Genetic Aberrations on Risk Stratification

Recent updates have incorporated genetic aberrations in prognostic scoring systems. Because several genetic aberrations associated with prognosis were identified in this study, we tried to evaluate their contribution in refining risk stratification. To evaluate the impact of genetic aberrations on risk stratification, we compared the risk of each patient group before and after applying genetic aberrations. Although there were statistically significant relationships between risk groups (contingence coefficient 0.514–0.554), we found some interesting changes. Thirty-seven of the 55 PV patients had more than one risk factor, including advanced age, leukocytosis and a vascular event [17]. When adjusting for MIPSS-PV (which included *SRSF2* mutation), 20 patients (54.1%) were reclassified into the low risk group. In ET, 28 patients had more than one risk factor, including advanced age, leukocytosis and a vascular event [18]. When applying MIPSS-ET (which included *SF3B1*, *SRSF2*, *TP53* and *U2AF1* mutations), the majority of patients (67.9%, 19/28) were reclassified as being low risk (Table 4). Among patients with PMF, 25 were classified as DIPSS low risk, 16 as intermediate risk, 21 as intermediate-2, and 4 as high risk (*n* = 4). It is notable that 10 patients and 6 patients in the low risk group were reclassified into intermediate and high risk groups based on MIPSS70 [22] and MIPSS70+ [34], respectively. MIPSS70 includes *ASXL1*, *EZH2*, *SRSF2* and *IDH1/2* mutations, while MIPSS70+ includes *ASXL1*, *EZH2*, *SRSF2* and *IDH1/2* mutations and cytogenetic risk categories. Three patients in the intermediate-1 risk group were reclassified into high and very high risk groups by MIPSS70 and MIPSS70+, respectively (Table 5). This meant that the newly developed risk stratification systems that include genetic aberrations discriminated more patients with low risk in PV and ET. On the other hand, the new systems selected PMF patients with higher risk among those with low or intermediate-1 risk.

Additionally, we analyzed the prediction capacity of mutation-enhanced prognostic scoring systems. In ET, MIPSS-ET revealed a significant prediction of OS (*P* = 0.003), while the previous risk factors did not (*P* = 0.922). In PMF, MIPSS70+ (*P* = 0.003) and MIPSS70 (*P* = 0.006) had a lower *P* value for the prediction of OS than DIPSS (*P* = 0.024). MIPSS70+ (*P* = 0.002) and MIPSS70 (*P* = 0.005) predicted EFS significantly better than did DIPSS (*P* = 0.201) (Appendix A). These findings indicated that the incorporation of NGS results improved prognostication, as the mutation-enhanced prognostic models predicted OS in a statistically significant manner.

## 4. Discussion

Continuous advances in genetic technology have resulted in great changes in clinical practice. As a consequence of these advances, current diagnoses and classifications of hematologic malignancies are largely based on genetic aberrations. The diagnosis of MPNs is made primarily based on clinical and hematologic features supported by mutations in *JAK2*, *CALR* and/or *MPL* genes. In the absence of mutations in any of these three genes, other myeloid neoplasm-associated mutations are recognized as other clonal markers for ET and PMF [1].

In the current study, we identified the driver mutations in 76.4% of MPN patients, with the highest frequency in PV patients (89.1%), followed by ET (72.8%) and PMF (69.7%) patients. There are no clear definitions of myeloid neoplasm-associated mutations other than those in seven genes in the WHO classification. We found that gene mutations other than the three driver genes and cytogenetic abnormalities accounted for a considerable proportion of cases. When applying seven myeloid neoplasm-associated genes proposed by WHO classification, eight patients began to possess clonal genetic markers. When including all detectable genetic aberrations, 12 patients were added to the list of patients with clonal genetic markers, and the proportion of patients with clonal genetic markers increased by 10.0%. The added mutations and karyotype abnormalities were commonly detected in myeloid neoplasms or other malignancies, such as *U2AF1*, *DNMT3A*, *RUNX1*, *TP53*, del(20q) and −5/del(5q). Therefore, it is necessary to define specific gene mutations and karyotype abnormalities and to clarify their roles in the diagnosis of MPN.

Overall genetic landscapes were similar to those reported in previous studies [35,36]. We found significant co-occurring mutations. *JAK2/SF3B1* co-mutations, which are known to be associated with myelodysplastic/myeoproliferative neoplasms with ring sideroblasts and thrombocytosis (MDS/MPN-RS-T), were identified in six patients (two ET and four PMF) [37,38]. It is notable that ring sideroblasts (RSs) were detected in all available BM samples with *JAK2/SF3B1* co-mutations. Even though the proportion of RS was far less than 5%, it was evident that the *SF3B1* mutation affected the formation of RS. Further studies should determine an appropriate diagnostic strategy for this disease, which has a broad spectrum of phenotypes. *TP53* mutation, co-occurring with del(13q), −5/del(5q), −7/del(7q) and a complex karyotype, was a significant adverse survival factor. Studies have shown that *TP53* mutations occur at a low frequency in chronic-phase MPNs and increase significantly according to disease progression. *TP53* mutations were positively correlated with poor survival outcomes, including early death [12,13,29], and −7/del(7q) was associated with poor survival in PMF patients [39]. Because all patients with *TP53* mutations had a complex karyotype, we postulated that the presence of *TP53* mutation resulted from the accumulation of genetic aberrations through disease progression.

In addition, we identified genetic aberrations associated with disease progression. Results from this and previous studies showed that *IDH1/2* mutations are associated with a risk of leukemic transformation. *IDH1/2* mutations occurred frequently in the blast phase of MPN compared to the chronic phase, suggesting a pathogenic contribution to leukemic transformation [40]. This is worth noting because IDH inhibitors may offer a therapeutic advantage for high risk MPN patients [41]. We also found that *RUNX1* mutations were predictive of leukemic transformation. This finding is consistent with previous studies that demonstrated that *RUNX1* mutations had adverse impacts on leukemia-free survival in PV and ET patients [24,25] and that *RUNX1*-mutated PMF patients had inferior overall survival and leukemia-free survival [42]. Another important gene was *ASXL1*; mutations in this gene are typically associated with fibrotic progression. *ASXL1* was the most frequently mutated gene, with the exception of the three driver genes. Previous studies showed that *ASXL1* mutations were enriched in patients with PMF [36], and *ASXL1* mutations are considered to be high risk mutations along with mutations in epigenetic regulators, such as *SRSF2* and *EZH2* in patients with MPNs [16,25,35,43]. *TET2* is an epigenetic regulator frequently mutated in all MPN subtypes. Although previous studies have demonstrated that *TET2* mutations increase the self-renewal capacity of hematopoietic stem cells [44] and can lead to disease initiation and progression [14], the prognostic value of *TET2* mutations is controversial [15,45]. Our study revealed that *TET2* mutations were frequently detected in MPNs with little prognostic impact. Del(20q) was another significant factor associated with fibrosis progression. It is notable that most instances of del(20q) were identified in PMF patients or patients with PV or ET with fibrosis progression during follow-up.

Clinicians should stratify each patient’s risk on the basis of updated information. This is even more important in the current era due to the increased availability of therapeutic options. Recent updates included genetic abnormalities into risk stratification for MPNs; *SRSF2* for PV, *SF3B1*, *SRSF2*, *TP53* and *U2AF1* for ET and *ASXL1*, *EZH2*, *SRSF2*, *IDH1/2* and karyotype for PMF. It was expected that risk groups were correlated before and after applying genetic abnormalities; however, there were changes worth paying attention to. In PV and ET, mutation-enhanced risk stratification system appeared to select low risk patients among patients with clinical and laboratory risk variables. By contrast, in PMF, mutation and additional karyotype-enhanced risk stratification systems identified higher risk patients among low or intermediate-1 risk groups, as supported by a previous finding that demonstrated the significance of *ASXL1* and *SRSF2* mutations for treatment decision-making in low or intermediate-1 risk patients with PMF [46].

We used a customized myeloid panel, which included disease-associated mutations and prognostic factors. Various commercial and customized NGS assays have been introduced in clinic, and they cover most significant genes for diagnosis and risk stratification in this study [47,48]. Bioinformatics were also incorporated in commercial NGS assays and increase the liability of analyzed results. The implementation of the NGS assay in clinical laboratories has been accelerated by making guidelines for the validation of the assay including panel design, operating setup and bioinformatics pipeline [49,50]. MPNs highly benefitted from NGS with efficient recognition of the presence of clonal markers in triple-negative MPN patients as well as the driver mutations. Although running cost and turnaround time are still challenging, as NGS moves into clinic, it becomes closer to patients and it influences risk stratification and therapeutic planning. Because it could eventually decrease medical cost through optimizing precision medicine, public health insurance programs will expand their coverage for NGS [51].

Due to the restricted number and less tight enrollment criteria of patients in each disease category and the relatively short follow-up period, further studies should be performed in a larger cohort considering racial heterogeneity. However, this study highlights the diagnostic and potential prognostic usefulness of targeted sequencing in the NGS era. In addition, the results of our contemporary study can be used as a valuable source to tailor therapeutic plans. Integrated genetic information facilitates not only diagnosis and prognostic expectations, but also our understanding of the presentation and progression of MPNs.

## 5. Conclusions

The genetic landscape demonstrated that 86.4% of MPN patients harbored clonal markers, including 76.4% of MPN patients, who harbored triple driver mutations. Some genetic aberrations (such as *TP53*, *IDH1/2*, *SF3B1*, *RUNX1*, −7/del(7q) and del(20q)) were related to overall survival and/or disease progression, and mutation-enhanced scoring systems performed better than phenotype-based risk stratification. These findings support the conclusion that integrated genetic information has a significant impact on the diagnosis and prognostication of MPN patients.

## Figures and Tables

**Figure 1 jcm-10-01033-f001:**
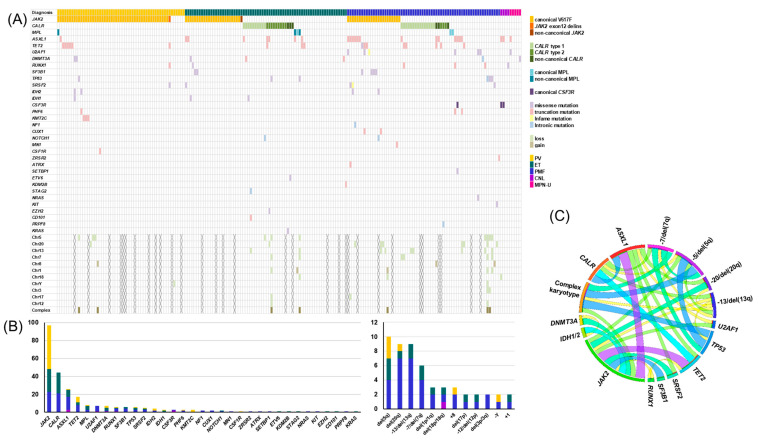
Genomic landscapes of 200 MPN patients. (**A**) The mosaic plot (**B**) The bar-charts of mutations and chromosomal abnormalities (**C**) The circos plot indicating recurrent co-occurrences of genetic aberrations by ribbon widths. PV: Polycythemia vera, ET: essential thrombocythemia, PMF: primary myelofibrosis, CNL: chronic neutrophilic leukemia, MPN-U: MPN-unclassifiable and MPN: myeloproliferative neoplasm.

**Figure 2 jcm-10-01033-f002:**
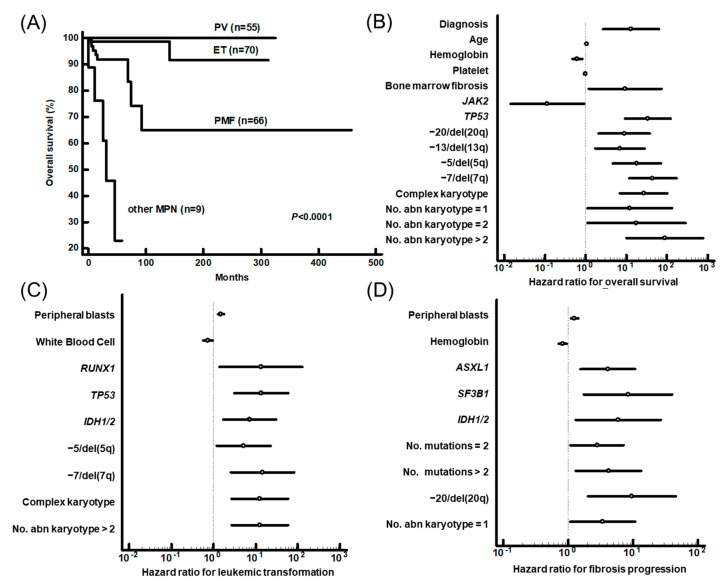
(**A**) Overall survival curves of the patients according to disease category. (**B**) Hazard ratios of significant factors for overall survival and (**C**) leukemic transformation in PV, ET, PMF and (**D**) fibrosis progression in PV and ET patients. PV: Polycythemia vera, ET: essential thrombocythemia, PMF: primary myelofibrosis, MPN: myeloproliferative neoplasm, No: number and abn: abnormal.

**Table 1 jcm-10-01033-t001:** Overall characteristics of patients in this study.

Variables	Total N = 200	PV N = 55	ET N = 70	PMF N = 66	Other MPN ^a^ N = 9	*p*
Age at diagnosis, mean ± SD	49.0 ± 14.7	50.3 ± 12.3	43.2 ± 15.3	52.5 ± 13.6	61.4 ± 14.2	<0.001
Sex, male (%)	43.0 (86/200)	52.7 (29/55)	35.7 (25/70)	39.4 (26/66)	66.7 (6/9)	0.109
Follow-up months, mean ± SD	64.7 ± 69.4	76.4 ± 67.7	78.5 ± 73.8	46.0 ± 64.8	23.6 ± 20.2	<0.001
White blood cells (10^9^/l), mean ± SD	12.9 ± 13.7	12.9 ± 7.4	9.5 ± 5.5	11.8 ± 11.9	47.8 ± 37.3	<0.001
Hemoglobin (g/l), mean ± SD	129.3 ± 33.6	164.7 ± 31.3	126.2 ± 20.6	107.7 ± 21.3	95.3 ± 18.7	<0.001
Platelet (10^9^/l), mean ± SD	560.5 ± 484.8	542.2 ± 324.9	803.7 ± 608.7	367.5 ± 340.3	197.2 ± 147.5	<0.001
Bone marrow fibrosis, %	46.0 (92/200)	16.4 (9/55)	21.4 (15/70)	100.0 (66/66)	22.2 (2/9)	0.064
Splenomegaly, %	31.5 (63/200)	27.3 (15/55)	11.4 (8/70)	54.5 (36/66)	44.4 (4/9)	0.364
Vascular event ^b^, %	20.0 (40/200)	36.4 (20/55)	20.0 (14/70)	9.1 (6/66)	0 (0/9)	0.275
Abnormal karyotype ^c^, %	22.5 (38/169)	18.6 (8/43)	9.8 (6/61)	37.5 (21/56)	33.3 (3/9)	0.003
Complex karyotype ^c^, %	4.7 (8/169)	4.7 (2/43)	3.3 (2/61)	7.1 (4/56)	0 (0/9)	0.692
Number of mutations, mean ± SD	1.3 ± 1.0	1.4 ± 0.9	1.1 ± 0.9	1.6 ± 1.1	0.9 ± 0.9	0.024

PV: polycythemia vera, ET: essential thrombocythemia, PMF: primary myelofibrosis, MPN: myeloproliferative neoplasm, CNL: chronic neutrophilic leukemia and SD: standard deviation. Group comparisons were performed using the Kruskal–Wallis and Chi-square tests for continuous and categorical variables, respectively; ^a^ Other MPN include CNL (*n* = 4) and MPN-unclassifiable (*n* = 5); ^b^ Vascular events include a thrombosis history at diagnosis and related microcirculation symptoms such as headaches, lightheadedness, atypical chest pain, transient visual disturbances and acral paresthesia; ^c^ Total of 169 BM karyotypes were available.

**Table 2 jcm-10-01033-t002:** Proportion of patients possessing a clonal genetic marker after applying sequential criteria for genetic aberrations.

	PV	ET	PMF	3 MPNs ^a^	Other MPN ^b^
Case number	55	70	66	191	9
Triple mutations ^c^	49 (89.1%)	51 (72.9%)	46 (69.7%)	146 (76.4%)	2 (22.2%)
Any of the seven mutations ^d^	0 (0%)	2 (2.9%)	6 (9.1%)	8 (4.2%)	3 (33.3%)
Other mutations ^e^ and/or abnormal karyotypes	1 (1.8%)	1 (1.4%)	9 (13.6%)	11 (5.8%)	4 (22.4%)
Any clonal genetic marker ^f^	50 (90.9%)	54 (77.1%)	61 (92.4%)	165 (86.4%)	7 (77.8%)
All negative	5 (9.1%)	16 (22.9%)	5 (7.6%)	26 (13.6%)	2 (22.2%)

PV: polycythemia vera, ET: essential thrombocythemia, PMF: primary myelofibrosis, MPN: myeloproliferative neoplasm and CNL: chronic neutrophilic leukemia. ^a^ PV, ET and PMF, ^b^ CNL (*n* = 4) and MPN-unclassifiable (*n* = 5), ^c^
*JAK2*, *CALR* and *MPL* for three MPNs and *CSF3R* for CNL, ^d^ any mutations including *ASXL1*, *EZH2*, *TET2*, *IDH1/2*, *SRSF2* and *SF3B1* among triple-negative patients, ^e^ any mutation other than the above-mentioned mutations, ^f^ any mutation and/or abnormal karyotype.

**Table 3 jcm-10-01033-t003:** Multivariate analysis of clinical and genetic factors for overall survival, leukemic transformation and fibrosis progression.

Variables	Overall Survival			Leukemic Transformation			Fibrotic Progression		
	*P*	HR	95% CI	*P*	HR	95% CI	*P*	HR	95% CI
Diagnosis ^a^	0.0087	78.2	3.0–2027.3						
PB blasts (%)				0.0486	1.3	1.0-1.6			
No. mutation							0.0352	2.0	1.1–4.0
***ASXL1***							0.0358	4.3	1.1–16.4
***RUNX1***				0.005	68.1	3.6–1300.4			
***SF3B1***							0.0009	31.5	4.1–243.3
***TP53***	0.0041	64.2	3.8–1096.5	0.0364	16.3	1.2–222.7			
***IDH1/2***				0.0051	32.5	2.8–371.1	0.0011	21.2	3.4–132.2
−7/del(7q)	0.0219	14.0	1.5–132.7						
del(20q)							0.0002	44.5	6.1–323.0

HR: hazard ratio, CI: confidence interval, PB: peripheral blood and No.: number. Variables assigned to be significant prognostic factors in univariate analysis were included in multivariate analysis. The fibrosis progression was analyzed among patients with polycythemia vera and essential thrombocythemia, excluding primary myelofibrosis. ^a^ Diagnosis of primary myelofibrosis.

**Table 4 jcm-10-01033-t004:** Comparison of risk groups in patients with polycythemia vera (PV) and essential thrombocythemia (ET).

Risk Group	Low	Intermediate	High	*P* (Contingency Coefficient)
MIPSS-PV	18	23	14	<0.001 (0.514)
Low	18	14	6	
Intermediate	0	9	5	
High	0	0	3	
MIPSS-ET	42	24	4	<0.001 (0.554)
Low	41	19	0	
Intermediate	1	3	2	
High	0	2	2	

MIPSS: Mutation-enhanced International Prognostic Scoring System, PV: polycythemia vera and ET: essential thrombocythemia. Risk factors for PV: advanced age (≥67 years, 5 points; 57–66 years, 2 points), leukocytosis (≥15 × 10^9^/l, 1 point), and the presence of a vascular event (1 point); low risk (no risk factors), intermediate risk (1–2 points) and high risk (≥3 points). Risk factors for ET: advanced age (≥60 years, 2 points), leukocytosis (≥11 × 10^9^/l, 1 point) and the presence of vascular event (1 point); low risk (no risk factors), intermediate risk (1–2 points) and high risk (3–4 points).

**Table 5 jcm-10-01033-t005:** Comparison of risk groups in primary myelofibrosis (PMF) patients.

DIPSS	Low	Int-1	Int-2	High	*P* (Contingency Coefficient)
MIPSS70	25	16	21	4	<0.001 (0.620)
Low	15	1	0	0	
Intermediate	8	12	11	0	
High	2	3	10	4	
MIPSS70+	21	13	19	3	<0.001 (0.603)
Very low	5	1	0	0	
Low	10	5	2	0	
Intermediate	4	4	3	0	
High	2	3	8	1	
Very High	0	0	6	2	

DIPSS: Dynamic International Prognostic Scoring System, Int: intermediate, MIPSS70: Mutation-enhanced International prognostic scoring system and MIPSS70+: Karyotype-enhanced MIPSS70.

## Data Availability

The data presented in this study are available on request from the corresponding author. The data are not publicly available due to ethical concern.

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
