# Peer review of "Impact of Integrated Genetic Information on Diagnosis and Prognostication for Myeloproliferative Neoplasms in the Next-Generation Sequencing Era"

_jcm, 2021, doi:10.3390/jcm10051033_

Round 1

Reviewer 1 Report

In this issue the author, integrate genetic alteration known to be found in myeloid malignancies to better inform and diagnostic and prognostic of myeloproliferative neoplasm. To do so they analyzed the genome landscape mutation of 200 patients with MPN.

They then evaluate the impact of MPN associated mutations on the diagnosis and risk stratification of disease. To the main three mutant driver (JAK2, CALR, MPL) and clonal genetic markers used in the WHO classification, the authors considered 9 gene mutations (DNMT3A, RUNX1, TP53, ZRSR2, KDM2B, 181 NRAS, KIT) and abnormal karyotypes (del(20q) and -5/del(5q)) as additional genetic markers.  By implementing all detectable genetic aberration, they improve the proportion of patients with clonal genetic markers up to 86.45% compare to 76.45% when done with the three main MPN driver. They also highlight genetic aberration associated with outcome such as inferior survival, leukemic transformation and fibrotic progression. Despite limited numbers of patients genetic aberrations identified as risk factors by multivariate analysis had statistical significance in log rank test. They then integrate this genetic aberration to see their impact on patient risk stratification. By comparing risk stratification systems the authors found that their mutation-enhanced prognostic system could identify lower risk Polycythemia Vera and Essential Thrombocythemia and higher risk Primary MyeloFibrosis. Importantly, the authors could improve the prediction of patient outcome using their new risk stratification scoring system.

The present study is of significant interest and relevant to MPN field of research. The rational and aim of the work is quite clear, the manuscript is well written and comprehensive. Methods used in the manuscript are appropriate and relevant to assess scientific problematic. The discussion is clear and comprehensive.

Manuscript status ; Accepted after Minor Revisions:

Minor change and add on should be done to improve the quality of the manuscript before publication

 Introduction

1- Introduction describe the research context but need to be improved

from Line 41 to 48: Reference are not sufficient and not cited in good position. This should be improve for clarity and relevance.

Along introduction the author go through MPN definition, subclass, driver mutation (occurrence and incidence on diagnosis), description of the scoring system used for risk stratification of MPNs.

Line 54- the authors introduce the risk stratification scoring system. Here it will be interesting to complete with a small paragraph or 1-2 sentences introducing the WHO classifications for MPNs (diagnosis, subtype and risk stratification scoring system). This will improve manuscript content and clarity. Further, because the WHO classification is later mention in the method and is a crucial factor in this study.

2- material and method

Line 73 :  “ Among patients diagnosed with Philadelphia-negative MPN”

During the introduction, the authors never mention MPN subtypes as Philadelphia-negative MPN. For consistency, they should define them the same way all along the manuscript.

Line 185: Typos 2 dots

3-Result

First, a global comment on the result part. There is missing conclusion at the end of each paragraph of these part of the manuscript (line 152,185,222,263). One or two sentences concluding on the finding and introducing next paragraph is necessary. This will improve quality and strength of the manuscript.

Figure 1: In legend, top right Corner. There is misspelling for the canonical JAK2 mutation. This should be modify to V617F

Table2: “Any clonal genetic markerf” Typos

Table 5: information are missing in table 5. The first line showing PMF patient number and respective risk classification should be presented the same way than in the in the table 4.

Supplementary material

Supplementary Figure 1: needs legend and annotation

Author Response

Thank you very much for your sincere and warm acknowledgment about our manuscript. We are delighted to hear that you think our work is impressive in our field. In the following sections, you will find our responses to each of your points and suggestions. We are grateful for the time and energy you have expended on our behalf.

Minor change and add on should be done to improve the quality of the manuscript before publication

 Introduction

1- Introduction describe the research context but need to be improved

from Line 41 to 48: Reference are not sufficient and not cited in good position. This should be improve for clarity and relevance.

  • We added more references in the appropriate positions.

Along introduction the author go through MPN definition, subclass, driver mutation (occurrence and incidence on diagnosis), description of the scoring system used for risk stratification of MPNs.Line 54- the authors introduce the risk stratification scoring system. Here it will be interesting to complete with a small paragraph or 1-2 sentences introducing the WHO classifications for MPNs (diagnosis, subtype and risk stratification scoring system). This will improve manuscript content and clarity. Further, because the WHO classification is later mention in the method and is a crucial factor in this study.

  • We added a few sentence about diagnosis of MPN by 2016 WHO classification system in lines 39-42. Subtypes and risk stratification scoring systems were introduced in lines 35-39 and 57-65.

2- material and method

Line 73 :  “ Among patients diagnosed with Philadelphia-negative MPN”During the introduction, the authors never mention MPN subtypes as Philadelphia-negative MPN. For consistency, they should define them the same way all along the manuscript.

  • We changed the “Philadelphia-negative MPN” to “MPN” for consistency of the manuscript.

Line 185: Typos 2 dots

  • The typo has been corrected.

3-Result

First, a global comment on the result part. There is missing conclusion at the end of each paragraph of these part of the manuscript (line 152,185,222,263). One or two sentences concluding on the finding and introducing next paragraph is necessary. This will improve quality and strength of the manuscript.

  • According to the reviewer’s suggestion, we added the concluding and introducing sentences in line 155-157, 189-191, 231-232.

Figure 1: In legend, top right Corner. There is misspelling for the canonical JAK2 mutation. This should be modify to V617F

  • We modified the error.

Table2: “Any clonal genetic markerf” Typos

  • The typo has been corrected.

Table 5: information are missing in table 5. The first line showing PMF patient number and respective risk classification should be presented the same way than in the in the table 4.

  • We added the numbers of patients for each risk classifications in the same way that presented in the table 4.

Supplementary material

Supplementary Figure 1: needs legend and annotation

  • We added the legend and annotation.

Reviewer 2 Report

Drs Lee and colleagues present the impact of applying next generation sequencing and mutation-enhanced prognostic models to their center's cohort of 200 patients with MPNs. The key findings were the following:

  • NGS testing strengthened diagnosis, especially for cases without mutations in JAK2, CALR, or MPL.
  • Co-occuring mutations affected risk of leukemic transformation or fibrotic progresison.
  • Incorporation of NGS results improved prognostication, as the mutation-enhanced prognostic models predicted OS in statistically significant manner in their cohort, whereas the clinical models generally did not. 

The methods and conclusions are sound.

I think these results are meaningful to publish. They demonstrate the utility of NGS in a "real world" setting and provide additional validation for the prognostic models.

Another strength of the work is the long (400 month) follow-up. Though I suppose the retrospective nature of the study leaves the question open of how the data guides treatment. In the future, one can determine whether this information can be applied prospectively to improve outcomes for high risk patients (eg: upfront use of allogeneic transplant).

The patients were selected based on those "who requested an NGS study" (section 2.1). Do the authors think this may bias the data in any way towards including more patients with either financial resources or perhaps better overall health?

Finally, though the number is small, it may be good to more specifically describe if the mutations correlated with prognosis for the CNL/MPN-unclassifiable group, as this had the worst outcomes. Did any mutations (or lack of mutation) correlate with a less aggressive course? Granted, with n=9, one may not be able to draw any conclusions from this cohort.

Author Response

The patients were selected based on those "who requested an NGS study" (section 2.1). Do the authors think this may bias the data in any way towards including more patients with either financial resources or perhaps better overall health?

  • Thank you very much for providing us with supportive and constructive reviews about our manuscript. The selection bias that you pointed out is worthy to discuss. The NGS panel test is supported by public health insurance program in Korea since Aug-2019. Therefore, the financial resources had less impact on the data in this study. We considered that more MPN patients presenting signs of disease progression were enrolled because clinicians tended to request NGS study before decision of aggressive treatment in a real world. Therefore, we mentioned about the less tight enrollment criteria with limitation of restricted number in the discussion section, in lines 377-379.

Finally, though the number is small, it may be good to more specifically describe if the mutations correlated with prognosis for the CNL/MPN-unclassifiable group, as this had the worst outcomes. Did any mutations (or lack of mutation) correlate with a less aggressive course? Granted, with n=9, one may not be able to draw any conclusions from this cohort.

  • Thank you for providing this direction. Per your suggestion, we have analyzed the factors correlated with OS in other MPN patients and found that patients with more genetic aberrations (≥ 2) showed poor outcome (P = 0.041). We added this results in revised manuscript, in lines 206-207.

Reviewer 3 Report

In the manuscript Dr. Lee and colleagues analyzed the genomic landscape of 200 patients with MPNs and evaluated the impact of genomic landscape on diagnosis and risk stratification in the clinical practice. 

Firstly they demonstrated that more than 90% of MPN patients carry at least a genetic abnormality, making this field particularly interesting and matter of debate in the clinical routine.

Moreover, they demonstrated that the "integrated" stratification systems  discriminated more patients with PV and ET in low risk subgroup, while a higher number of cases with PMF entered into a higher risk category. I think that this latest group and finding would be more underlined because this could have a clear consequence in the choice of alloBMT instead of the re-allocation of PV and ET cases in the low category (in these MPNs the impact of therapeutic choice is lower).

Thus, I suggest some minor revisions:

1: in the discussion section, it would be interesting to firstly discuss the possible reproducibility of this new scoring system in respect of the conventional few genes analysis in the routine until now: i.e., costs, eventual available commercial kits, needed expertise. It is necessary convincing the reader that it would be implemented in the clinical practice, with acceptable costs

2: the authors could try to apply their new genetic characterization to their series of patients using the web site from Sanger https://www.sanger.ac.uk/science/tools/progmod/progmod/

this could be an interesting comparator that is already employed in the clinical practice.

3. please consider the table 5: check the format and the real presence of numbers in the first raw

4. check summary: SF3B1 vs SF3B2?

Author Response

Thus, I suggest some minor revisions:

1: in the discussion section, it would be interesting to firstly discuss the possible reproducibility of this new scoring system in respect of the conventional few genes analysis in the routine until now: i.e., costs, eventual available commercial kits, needed expertise. It is necessary convincing the reader that it would be implemented in the clinical practice, with acceptable costs

  • We appreciate all your insightful comments. As the reviewer suggested, we discussed about the assessment of the NGS tests in the clinical practices as follows (in lines 364-376). We hope that you find these revisions an improvement.

2: the authors could try to apply their new genetic characterization to their series of patients using the web site from Sanger https://www.sanger.ac.uk/science/tools/progmod/progmod/

this could be an interesting comparator that is already employed in the clinical practice.

  • As you suggested, we have tried to apply the prognostic tool to our cases and found that the tool did not fully predict patient’s outcome. Two ET patients with 9 and 15 years expected median EFS died at 0.4 and 11.7 years. Six PMF patients with 5, 7, 9, 9, 10, 11 years expected median EFS died at 5, 1, 0.4, 6.2, 1.33 and 7.75 years after diagnosis. These discrepancies may be because of racial differences, however, we have saved the proper comparison for later due to the limitation of our study including the restricted number of patients and relatively short follow-up period.
  1. please consider the table 5: check the format and the real presence of numbers in the first raw
  • We added the numbers of patients for each risk classifications in the same way that presented in the table 4.
  1. check summary: SF3B1 vs SF3B2?
  • The authors corrected the typo.